# Physiotherapists’ Experiences with the Hip Disability and Knee Injury Osteoarthritis Outcome Score Following Total Hip and Knee Arthroplasty: A Qualitative Interview Study

**DOI:** 10.3390/jcm14030992

**Published:** 2025-02-04

**Authors:** Dennis J. van den Berg, Esther T. Maas, Rosa-Lynn Edelaar, Mathijs B. Arendsen, Elizabeth J. de Louw, Henri Kiers, Thea P. M. Vliet Vlieland, Raymond W. J. G. Ostelo, Marianne H. Donker

**Affiliations:** 1Department of Health Sciences, Faculty of Science, Vrije University Amsterdam, 1081 HV Amsterdam, The Netherlands; 2Amsterdam Movement Sciences Research Institute, 1081 HV Amsterdam, The Netherlands; 3Institute for Movement Studies, University of Applied Sciences Utrecht, 3584 CS Utrecht, The Netherlands; 4Dutch Association for Quality in Physiotherapy, 8031 DX Zwolle, The Netherlands; 5Department of Orthopaedics, Rehabilitation and Physiotherapy, Leiden University Medical Center, 2333 ZA Leiden, The Netherlands; 6Basalt Rehabilitation, 2333 AL Leiden, The Netherlands; 7Department of Health, University of Applied Sciences Leiden, 2333 CK Leiden, The Netherlands; 8Department of Epidemiology and Data Science, Amsterdam UMC, Location Vrije Universiteit, 1081 HV Amsterdam, The Netherlands

**Keywords:** total hip arthroplasty, total knee arthroplasty, PROMs, HOOS, KOOS, thematic analysis, physiotherapy

## Abstract

**Background:** Clinical guidelines for physiotherapy following total hip and knee arthroplasty (THA/TKA) recommend using Patient-Reported Outcome Measures (PROMs) like the Hip disability and Knee injury Osteoarthritis Outcome Score (HOOS/KOOS). However, it remains unclear how physiotherapists use these PROMs in daily clinical practice. **Objective:** To explore primary care physiotherapists’ experiences with the HOOS/KOOS in daily clinical practice following THA and TKA. **Methods:** Thirteen physiotherapists in the Netherlands were recruited via convenience sampling. Data were collected through semi-structured interviews, which explored HOOS/KOOS use in clinical practice, administrative regulations, and applications beyond patient care, as well as think-aloud interviews to capture perceptions of the content of these PROMs and interpretations of hypothetical patient scores. Interviews were analysed using reflexive thematic analysis. **Results:** The physiotherapists’ ages ranged from 25 to 54y, with annual THA/TKA volumes from 5 to 50 patients. Three themes emerged, as follows: (1) “Physiotherapists use the HOOS/KOOS for various purposes in daily clinical practice”, including complementing history taking and monitoring patient progress; (2) “Perceptions of the relevance of the HOOS/KOOS for daily clinical practice vary per item, domain, and version,” with items related to daily life activities and psychosocial factors being perceived as more valuable; and (3) “Practical aspects of HOOS/KOOS administration influence their use in daily clinical practice,” with electronic health records (EHRs) that facilitate PROM administration enhancing their use, while administrative regulations limit this. **Conclusions:** Physiotherapists experience HOOS/KOOS items related to daily life activities and psychosocial factors useful for history-taking and monitoring patient progress, particularly when embedded in EHRs that facilitate PROM administration.

## 1. Introduction

Value-based healthcare aims to maximise the value of care for patients and reduce the costs of healthcare [1]. A key component of value-based healthcare is the use of Patient-Reported Outcome Measures (PROMs) [2], which capture the patient’s own perspective of their health status [3]. PROMs can enhance communication between healthcare providers and patients, help to prioritise treatment goals in shared decision-making [2,4], and assess the achievement of goals from the patient’s perspective [5]. Additionally, value-based healthcare advocates to use PROMs for internal quality improvement and external quality information and transparency [6].

PROMs have become “key in demonstrating the success of physiotherapy” [7], and their integration into daily clinical practice has been shown to increase patient outcomes in neck and low back pain [8]. Treatment of patients following total hip and knee arthroplasty (THA/TKA) is very common in physiotherapy practice, and PROMs are recommended in various national and international physiotherapy guidelines [9,10,11,12,13,14]. Of these PROMs, the Hip disability and Knee injury Osteoarthritis Outcome Scores (HOOS/KOOS) are the most frequently recommended, both in national [11,13,14] and international [12,15] guidelines.

Regarding their actual usage in daily physiotherapy practice, knowledge is scarce, with only a few studies showing the use of PROMs by physiotherapists following THA and TKA being available [16,17,18]. In two survey studies among physiotherapists in Canada [16] and the United States [17], it was found that up to 59% and 70% of physiotherapists, respectively, used the Lower Extremity Functional scale, a PROM recommended by North American guidelines [9], for clinical decision-making. A Dutch survey study showed that less than 30% of physiotherapists used the HOOS or KOOS, which are recommended in the national physiotherapy guidelines for hip or knee osteoarthritis for clinical decision-making [18]. While these studies offer valuable insights, surveys lack the depth to explore the complexity of the use of PROMs in daily clinical practice, including how and why they are used in patient–therapist communication, shared-decision making, and progress evaluation. Qualitative methods, such as interviews [19], may provide deeper insights into physiotherapists’ experiences with PROMs, which can lead to a more comprehensive understanding of PROMs use in daily clinical practice. For physiotherapy following THA and TKA, the one qualitative study describing physiotherapists’ perceptions on best practices following THA and TKA elaborated on perceptions regarding objective measures, such as joint range of motion, but did not further explore experiences with the use of PROMs [20].

Concerning physiotherapists’ use of PROMs in other conditions, two studies utilised mixed methods to identify barriers and facilitators to PROMs use [21,22]. Common barriers included physical therapists’ competence, problems in changing behaviour, practice organisation (e.g., no time) and the poor availability and feasibility of PROMs [21,22]. However, these studies lacked detailed descriptions of physiotherapists’ experiences with PROMs. Such descriptions may complement established barriers and facilitators by providing a deeper, more personal understanding of the day-to-day realities of PROM usage. We identified only one study that described physiotherapists’ experiences with PROMs in detail [23]. That study showed that physiotherapists in Sweden felt that PROMs provided additional information, improved patient interaction, helped to set treatment goals, and monitored treatment progress in patients with low back pain [23]. However, the study encompassed multiple PROMs; arguably, focusing on a single PROM might allow for a more thorough exploration of their use in daily clinical practice.

Given that the HOOS and KOOS are recommended by both national [11,13,14] and international [12,15] physiotherapy guidelines for clinical practice, and that qualitative studies regarding physiotherapists’ experiences with the usage of these PROMs in daily clinical practice are lacking, this study aimed to explore the experiences of primary care physiotherapists with the use of the HOOS and KOOS in daily clinical practice following THA and TKA in the Netherlands. This study may increase understanding of the day-to-day realities of PROMs use in individual patient care following THA and TKA, and provide valuable insights for the development, adaptation, and implementation of PROMs as intended in value-based healthcare.

## 2. Materials and Methods

### 2.1. Study Design

We used an interpretative approach [24] to explore physiotherapists’ experiences with the HOOS and KOOS in individual patient care following THA and TKA. Data were collected through semi-structured [25] and think-aloud [26] interviews in April and May 2024, following the COREQ checklist for reporting [27].

### 2.2. Sampling

Physiotherapists in primary care with experience in treating patients following THA and TKA were recruited using a convenience sampling strategy. Recruitment occurred through the Dutch Association for Quality in Physical Therapy, via email invitation; through a professional collaboration network focusing on post-THA and TKA, via verbal invitation; and through the author’s personal networks. Additionally, students from the master’s programs in Physiotherapy at HU University of Applied Sciences Utrecht and the Musculoskeletal Physiotherapy Sciences and Human Movement Sciences Master’s programs at Vrije Universiteit Amsterdam were invited via email and verbal invitation. The eligibility criteria were as follows:Employment in a primary care setting;Experience with at least five THA/TKA patients per year;Use of the HOOS or KOOS, or their variants, following THA or TKA.

Eligible physiotherapists interested in participating in the study received an information letter via email. We aimed to include 12 physiotherapists, as a sample of 6 to 12 interviews is considered adequate [28].

### 2.3. The HOOS/KOOS

The Dutch guideline for hip and knee osteoarthritis recommends using the HOOS/KOOS or their shorter versions (HOOS-/KOOS-PS and HOOS-/KOOS-ADL) for post-joint replacement physiotherapy [13]. The KOOS [29] (42 items) and HOOS [30] (40 items) assess pain, symptoms, activities of daily living (ADL), sports and recreation, and quality of life, and are validated for TKA [31] and THA [30] patients. The HOOS-/KOOS-PS [32,33] (5–7 items) focus on ADL and sports and recreation, and the HOOS-/KOOS-ADL [29,34] (17 items) on ADL. Additionally, a 12-item version (HOOS-/KOOS-12 [35]) focusing on pain, ADL, and quality of life is not included in the guidelines.

### 2.4. Data Generation

Interviews were conducted in April and May 2024 using semi-structured interviews [25] for insights into physiotherapists’ experiences with the HOOS/KOOS, and think-aloud interviews to capture interpretations of hypothetical patients’ HOOS/KOOS scores and perceptions of the content of these PROMs. Two female bachelor’s students in health sciences (RE and EL) at Vrije Universiteit Amsterdam, who had no previous relationship with the participants, conducted the interviews. Both students had prior training as part of the health sciences curriculum and received extra training provided by the last author (MD), an experienced qualitative researcher.

#### 2.4.1. Semi-Structured Interviews

Interviews were conducted in-person in the participant’s work environment when possible; otherwise, they were held online via Microsoft Teams (version 24074.2607.2799.9843). A topic guide was developed based on a previous survey study [18], the Dutch guidelines for hip and knee osteoarthritis [18], and the PROM toolbox of the Dutch National Healthcare Institute [6]. The topic guide was pilot tested on the first author (DB), an experienced clinician meeting the eligibility criteria. After three interviews, the guide was revised to exclude a topic about physiotherapy care following THA and TKA, allowing for more focus on HOOS/KOOS usage. Questions related to the three remaining topics—‘experiences with HOOS/KOOS in clinical practice’, ‘regulations on HOOS/KOOS administration’, and ‘use of HOOS/KOOS outside patient care’—were refined iteratively during the data collection process. The finalised topic guide is provided in Appendix A.

#### 2.4.2. Think-Aloud

The think-aloud interviews took place at participants’ workplaces. Participants were shown baseline KOOS scores for a hypothetical 65-year-old male with TKA and baseline HOOS scores for a hypothetical 82-year-old female with THA. They were asked to think aloud while interpreting these scores and their clinical value. Follow-up scores for both patients were then presented, and participants again verbalised their thoughts. Finally, items from each HOOS/KOOS domain were shown, and participants were instructed to think aloud while evaluating their clinical value for THA/TKA patients. After each task, the interviewer used focused questions to clarify or supplement the data. An overview of the materials used is provided in Appendix B.

Both semi-structured and think-aloud interviews were audio-recorded and transcribed by the interviewers, after which recordings were deleted. Prior to the interviews, the interviewers informed participants about their occupation and obtained written informed consent. At the end of the interviews, the participant’s characteristics, work experience, highest educational level, clinical specialisation, annual volume of THA and TKA treatments, and currently used HOOS/KOOS versions were questioned. No other people, besides the interviewer and participant, were present during the interviews, which lasted approximately 30–60 min. Participants received a summary of the interview (member check) to allow them to correct misinterpretations and/or provide missing information.

### 2.5. Data Analysis

Interviews were analysed using reflexive thematic analysis, as described by Brown and Clark [24]. Analysis started after the first interview and took place in parallel with data collection, so that questions could be further developed throughout the data collection process to gain deeper insights into physiotherapists’ experiences with the HOOS and KOOS. After each interview, RE and EL made field notes which were discussed with the research group. The first interviews were coded independently by RE (semi-structured interviews) and EL (think-aloud interviews). Subsequently, the research team deliberated on these codes to ensure that multiple options and viewpoints were considered. RE (semi-structured interviews) and EL (think-aloud interviews) independently coded the remaining interviews. In second instance, DB independently coded both the semi-structured interviews and think-aloud interviews, resulting in two rounds of coding. All codes were collated, and initial patterns were recognized by DB and EM and further developed and reviewed in collaboration with MD. From these patterns, the team further developed themes by checking them against the coded data and the entire dataset to ensure that the themes reflected a convincing story of the data. Subsequently, the research team discussed each theme to determine its scope, focus, and appropriate naming. Finally, the themes were written down and illustrated with participants’ quotes, which required further refinement of themes and the revisiting of earlier phases to ensure that ideas still represented the data. In this phase, the analysis was also contextualised in relation to the existing literature, which helped to synthesize the analytical narrative into a meaningful story.

## 3. Results

### 3.1. Participants

Twenty-two physiotherapists responded to the study invitation, of whom thirteen were eligible. Six were recruited via the professional collaboration network, six via the Dutch Association for Quality in Physical Therapy, and one via the first author’s (DB) personal network. Seven participants were interviewed using semi-structured interviews, and six using think-aloud interviews. Participants’ ages ranged between 25 and 54 years, with 3 to 25 years of experience as a physiotherapist. Six participants were female, and all but two participants had a yearly THA or TKA treatment volume exceeding six patients. Specialities included manual therapy, sports physiotherapy, oedema therapy, and geriatric physiotherapy. A detailed overview of participant demographics is provided in Table 1.

### 3.2. Themes

Analysis of the data resulted in three main themes and nine subthemes: (1) physiotherapists use the HOOS and KOOS for various purposes in daily clinical practice; (2) perceptions of HOOS and KOOS relevance for daily clinical practice vary per item, domain, and version; and (3) practical aspects of HOOS and KOOS administration influence their use in daily clinical practice. Data saturation was reached after eight interviews, as no new themes emerged. Each theme is discussed in detail below and supported by quotes to illustrate key aspects of each (sub)theme. A full overview of themes and subthemes is provided in Table 2.

#### 3.2.1. Theme 1: Physiotherapists Use the HOOS and KOOS for Various Purposes in Daily Clinical Practice

Theme 1 outlines that physiotherapists use the HOOS and KOOS for different purposes, as follows: (1) to complement history taking and identify complications, and (2) to help monitor progress and set treatment goals.

##### The HOOS and KOOS Complement History Taking, Physical Examination, and Identifying Complications

The physiotherapists mostly use history taking, observation, and physical examination to examine the patient’s status. Some physiotherapists indicate that the HOOS and KOOS complement this process by identifying symptoms and limitations that have not come up during history taking or physical examination, thereby capturing potential important symptoms and activity limitations: *“Normally, you do assess the limitations, but the patient only mentions walking. However, stair climbing and other activities might also be problematic, but you don’t always have the time to explore these further.”* (P9). Low scores on the HOOS and KOOS domains initiate follow-up questions on the underlying stories behind these low scores and heighten physiotherapists’ alertness regarding possible complications: *“You can clearly see that symptoms like stiffness and pain are quite prominent. So, alarm bells start ringing, and you dive deeper into it, asking why this is happening and why they responded that way”* (P8). Additionally, one physiotherapist uses the HOOS and KOOS to check how patients perceive their limitations, as sometimes the perceptions of the physiotherapist do not align with HOOS and KOOS outcomes. *“We might think, “Oh, they’re doing fine,” but if you probe further or revisit a question, you might find they’ve lost all confidence in their knee*.*”* (P11).

##### The HOOS and KOOS Help to Monitor Progress and Set Treatment Goals

Some physiotherapists stated that individual HOOS and KOOS items are discussed with the patient to decide whether they should be part of the treatment goals: *“Shall we work on that (HOOS/KOOS item) then?” And sometimes someone says, “Yes, but why should I work on that? I don’t actually want to”* (P5). Many physiotherapists indicated, regarding the HOOS and KOOS outcomes, that *“It sometimes helps people realize they are making progress, even if they feel like they’re stuck or things are only getting worse”* (P3). Repeated administration of HOOS or KOOS also helps to identify deviations from the expected course: *“For example, the second time you administer it (HOOS/KOOS), you notice progress. But then, the third time, it worsens, and you think, “Okay, this is a strange pattern, what’s going on?”* (P2). Physiotherapists appear to use their experience as a reference for the expected course: *“Daily functioning is somewhat reduced, which is surprising, because normally after six weeks, you should be able to do more”* (P8). Additionally, changes or the lack of in HOOS and KOOS scores over time help to focus follow-up treatment: *“Or if there’s little progress, we might need to focus more on that (HOOS/KOOS item), if that’s what the patient also wants”* (P1).

#### 3.2.2. Theme 2: Perceptions of HOOS and KOOS Relevance for Daily Clinical Practice Varies per Item, Domain, and Version

Theme 2 outlines the varying perceptions among physiotherapists regarding the relevance of the content of the HOOS and KOOS for daily clinical practice in patients with THA or TKA. These perceptions differed in terms of that (1) the activities of daily life (ADL) subdomain supports clinical reasoning, whereas (2) the items of the pain, symptoms, and sports and recreation domains are unclear or irrelevant for most patients with a THA or TKA, or (3) have little clinical value. Consequently, (4) the HOOS/KOOS-ADL and HOOS/KOOS-12 are preferred above the full version of the HOOS/KOOS and the HOOS/KOOS-PS.

##### Items Regarding ADL Activities and Psychosocial Factors Support Clinical Reasoning

The subdomain “activities of the daily life” is valued because it contains items that are treatable by a physiotherapist and align with general treatment goals following THA and TKA: *“Yes, I often skim through these (items ADL subdomain). Where are they really struggling? If it’s specifically stair climbing, you can train that effectively. And it could also be a strength issue if they still can’t do it. A quick scan can guide the treatment direction”* (P12). The quality of life domain was also perceived as important by most physiotherapists. However, those physiotherapists also stated that *“the quality-of-life items don’t really capture quality of life, but focus more on functional recovery”* (P9). Some physiotherapists argued that the ADL domain is also representative of the quality of life domain: “ *And actually, those ADL items reflect a form of quality of life*.” (P8). Additionally, physiotherapists value items that address psychosocial issues which are harder to address in a conversation with the patient or are not easily seen during patient observation or physical examination: *“How much confidence do you have in your hips? Yes, that’s important because people don’t often mention it, but it’s frequently a concern.”* (P13).

##### Items of the Pain, Symptoms, and Sports and Recreation Domains Are Unclear or Irrelevant for Most Patients with a THA or TKA

Many physiotherapists stated that several items from the pain, symptoms, and sports and recreation function domains are unclear to patients or do not apply to all patients undergoing THA or TKA. Therefore, respondents question the validity of the HOOS and KOOS for a THA and TKA population: *“For example, there’s a question about whether someone can run, and well… most people with those complaints can’t. So, you could question how valid and representative these questionnaires are for the people we are using them for”* (P10). Consequently, respondents feel that these questions undermine their professional authority, leading them to fill out these items themselves: *“Do I really need to ask someone who’s just had surgery, ‘Can you run a bit?’ Yes, but do I really need to ask that? … These are no-brainers where you just know the answer is, ’No, they can’t.”* (P3). Moreover, non-applicable items can make patients insecure and “*It can also cause anxiety for people. Like, ‘Oh, can’t I do that yet?’ or ‘I’m feeling that, is that good or bad?*’’ (P9). In contrast, one physiotherapist indicated that those items increase questionnaire sensitivity: *“And someone with knee pain who can squat—there aren’t many of those. That’s why it’s a very sensitive question on your questionnaire. And that’s why I don’t always ask it during an assessment.”*

##### The Pain, Symptoms, and Sport and Recreation Domains Have Little Clinical Value

Many physiotherapists indicated that the pain and symptoms domain are important. However, most of these physiotherapists indicated that items within these domains *“are questions that I usually already ask during the conversation”* (P5). Additionally, some physiotherapists indicated that several items of the symptom’s domain are part of their physical examination or are observed during treatment: *“On the other hand, it (HOOS/KOOS symptoms domain) gives us a bit of information about whether they’re walking upright or have limitations in moving backward. But again, you notice that very quickly in practice”* (P11). Some respondents stated that the sports and recreation functioning domain is better suited for patient history taking than a standardized questionnaire, as sports and recreational activities *“just varies greatly between patients. So if one has this issue and the other something completely different, it’s quite difficult to encompass that in a single domain”* (P9).

##### Physiotherapists Prefer the HOOS/KOOS-ADL and HOOS/KOOS-12 over the Full Version of the HOOS/KOOS and the HOOS/KOOS-PS

Most respondents indicated that “*the HOOS and KOOS are far too long for us. They’re too burdensome for both the patient and the therapist, so we don’t find them functionally easy to apply*” (P3). Additionally, these longer versions, as well as the HOOS/KOOS-PS, are not perceived as suitable for post-THA and TKA physiotherapy care because the items do not align with this population, resulting in non-representative scores. The HOOS/KOOS-ADL and HOOS/KOOS-12 are deemed valuable because items within these versions are “*simple, short, and to the point, easy to answer*” (P6), and align with general treatment goals and patient interests: “*With a procedure like a new knee or hip, you want to ensure that people return to their daily lives and ADL activities with as little difficulty as possible. And the list, especially the twelve-item version, provides a concise overview where I can quickly see what treatable factors remain, where a person still has complaints, or is struggling, and how we’re going to solve that.*” (P4).

#### 3.2.3. Theme 3: Practical Aspects of HOOS and KOOS Administration Influence Their Use in Daily Clinical Practice

Theme 3 highlights the practical aspects of HOOS and KOOS administration that seem to impact their utilisation. The use of (1) a supportive electronic health record system appears to facilitate the use of the HOOS and KOOS. Conversely, (2) regulations regarding HOOS and KOOS administration imposed by insurance companies and professional associations have conflicting consequences. Additionally, (3) shorter more practical generic PROMs are preferred over the HOOS and KOOS by some physiotherapists.

##### Supportive Electronic Health Record Systems Facilitate the Use of HOOS and KOOS

Electronic health record systems that facilitate the administration of the HOOS and KOOS before the initial consultation seem to enhance the questionnaire’s integration into treatment. Respondents using a supportive electronic health record system indicated that it increases efficiency because it streamlines questionnaire administration, supports focused patient history taking, and saves time: *“What’s nice about sending it digitally is that people see filling out the questionnaires beforehand as part of their treatment time. Although they also see it as treatment time if done during the session, they then feel it’s a waste of time”* (P13). Additionally, electronic health record protocols appear to be decisive in determining which version of the HOOS or KOOS is employed: *“I didn’t even know there was a longer version. You just work with the computer system where our records are kept. And if you get someone for a total hip or knee replacement, there’s a certain protocol that indicates which tools to use. At our practice, the HOOS-PS and KOOS-PS are recommended*.*”* (P1).

##### Regulations Regarding HOOS and KOOS Administration Imposed by Insurance Companies and Professional Associations Have Conflicting Consequences

Many respondents indicated that the imposed regulations by health insurance companies or professional associations, along with financial incentives, were important reasons for administering HOOS and KOOS. However, physiotherapists resisted these regulations, believing they unnecessarily burdened patients and led to more therapists filling out PROMs themselves: *“And what you also see is that physiotherapists just fill in part of the questionnaires themselves because they think it’ll take too long with the patient, or it’s too burdensome”* (P1). Additionally, current regulations regarding the timing of HOOS and KOOS during the first consultation are not perceived as useful for patient care: *“In the first phase, patients can’t really do anything, and they struggle with everything. So, if I administer it then, I get the same score everywhere. It doesn’t add much value at that point, but after six weeks, it does”* (P4). Reports from professional associations on collected data, which present only the percentages of administered questionnaires, were not valued by physiotherapists: *“When you get a quarterly report back, showing your overall HOOS/KOOS usage in percentages… well, I don’t find that useful at all”* (P3). Moreover, physiotherapists reported being either unaware of or lacking the time to analyse HOOS and KOOS scores at the group level for internal quality improvement or external transparency.

##### Shorter, More Practical, and Generic PROMs Are Preferred over the HOOS and KOOS

Physiotherapists preferred the Patient Specific Complaints (PSC) questionnaire and the Numeric Pain Rating Scale (NPRS) for practical reasons, such as their length and ease of integration into patient history taking, especially during consultation at the patient’s home: *“But I do ask about the PSC and NPRS. That’s pain scoring and setting goals at the activity level because it’s easier to inquire about*.*”* (P1). Additionally, most physiotherapists indicated that the PSC is more suitable for deciding on treatment goals, because unlike the HOOS and KOOS, *“with a PSC, the client can indicate what their biggest problem is in daily functioning.”* (P7).

## 4. Discussion

### 4.1. Main Findings

This study aimed to explore the experiences of primary care physiotherapists with the use of the HOOS and KOOS in daily clinical practice following THA and TKA in the Netherlands. Three key themes emerged: First, physiotherapists reported using these questionnaires to complement history taking and physical examination, to monitor patient progress, and to set treatment goals. We observed that physiotherapists primarily relied on their own experience in interpreting HOOS and KOOS scores for these purposes. Second, most items within the pain, symptoms, and sports and recreation domains were perceived as irrelevant or having less clinical value. Items within the ADL domain and those capturing psychosocial factors were valued more because they aligned with treatment goals or were harder to address in conversation. Consequently, physiotherapists preferred the HOOS/KOOS-ADL and HOOS/KOOS-12 over the full HOOS/KOOS and the shorter HOOS-/KOOS-PS. Third, practical aspects of PROM administration influenced HOOS and KOOS use. Electronic health record systems facilitating PROM administration increased HOOS and KOOS use, while administrative regulations limited their use. Additionally, shorter, generic, single-question PROMs were preferred because physiotherapists found these PROMs easier to administer during history taking. We observed that physiotherapists utilizing supportive electronic health record systems valued the HOOS and KOOS for more purposes and were less critical of regulations regarding PROM administration. Additionally, we observed tension between regulatory requirements and the clinical context, as physiotherapists completed PROMs on behalf of the patient to preserve professional autonomy and minimize patient burden.

### 4.2. Comparison to the Literature

Only one other study described physiotherapist experiences with PROMs (in patients with low back pain) and also showed that physiotherapists used PROMs for various purposes [23]. Similar to our results, those physiotherapists used specific PROMs to complement history taking and physical examination and to evaluate progress [23]. However, those physiotherapists did not use such PROMs to help set treatment goals or identify complications, but instead used the more general PCS to set treatment goals [23], like most physiotherapists in our study.

Consistent with our results, physiotherapists in Sweden also experienced that patients find questions regarding psychosocial factors easier to answer in a PROM than in conversation [23]. Furthermore, several other studies reported that physiotherapists often experienced that the content of PROMs was misaligned with patient-specific goals or unapplicable for the target population [22,23,36,37], which was confirmed by our results for several items and domains of the HOOS and KOOS. Our study adds that for physiotherapy care following THA and TKA, HOOS/KOOS items regarding ADL align well with patient-specific goals. This alignment appeared to facilitate their integration into progress evaluation.

Regarding practical aspects of PROM administration, previous studies also showed that both physiotherapists [23] and patients [38] experienced increased treatment efficacy with electronic health record systems that facilitated PROM administration prior to consultation. However, our results suggest that this has not yet been fully implemented. Additionally, a previous study described the completion of PROMs by healthcare professionals themselves, due to tensions between regulations and the clinical context, as the concept of “gaming” [39].

This study highlights a discrepancy between HOOS and KOOS usage and the principles of value-based healthcare. We observed that physiotherapists relied on their own interpretations of HOOS and KOOS scores and mainly used within-patient changes to evaluate progress. This reliance may come from the absence of normative reference scores in the Dutch physiotherapy guidelines [13]. Normative reference scores are considered essential in value-based healthcare as they enable the comparison of individual patient scores to these reference values, which can help to determine whether treatment is likely to yield clinically meaningful benefits [2]. Without such comparisons, there may be a risk of under- or overtreatment, which can contribute to a variation in patient care.

PROMs can also contribute to value-based healthcare by integrating them into shared decision-making [4]. However, this process requires patients to select and prioritise items as treatment goals [4]. Our findings suggest that physiotherapists favoured the PSC for setting treatment goals, as it allowed patients to prioritise items—a feature lacking in the HOOS and KOOS. This could explain physiotherapists’ limited use of the HOOS and KOOS for shared decision-making [18].

### 4.3. Strengths and Limitations

This study’s strength lies in the use of both semi-structured and think-aloud interviews to gather rich and in-depth data on physiotherapists’ experiences with the HOOS and KOOS. Combining these methods provided a comprehensive understanding of both general and specific experiences of the HOOS and KOOS, with think-aloud data offering specific interpretations that supported theme development from interview data.

Arguably, the lack of clinical background and the relative inexperience of the interviewers might have constrained the depth of data collection, thereby limiting the credibility. However, this helped avoid biases from preconceived notions or assumptions about PROMs use in clinical care, thereby increasing dependability. Furthermore, to maintain credibility, we developed the topic guide with recognized guidelines [6,13] and used member checks, peer debriefing, and a thorough audit trail. Additionally, the study’s focus on the HOOS and KOOS in the context of THA and TKA, while allowing for an in-depth exploration of their use within this context, limits the applicability of the findings. Lastly, the convenience sample may imply that the physiotherapists have more affinity with PROMs use in daily clinical practice, which, in addition to the limited geographic scope, restricts the transferability of the results.

### 4.4. Implications for Clinical Practice and Research

Our results highlight the need for shorter PROMs that capture ADL activities, which align well with patient goals, as well as psychosocial factors, which are more challenging to address in conversation or are not easily observed. Thus, existing clinical practice guidelines for physiotherapy following THA and TKA should consider revising their recommendations to prioritise the HOOS/KOOS-ADL and the HOOS/KOOS-12. To facilitate their effective integration into daily clinical practice, primary care practices should consider adopting electronic health record systems that support PROM administration before consultation. Moreover, integrating normative reference values into clinical practice guidelines could further improve the integration of PROMs in daily clinical practice.

Future studies should explore how individual patient scores can be interpreted in relation to normative reference scores and their implications for treatment decisions. Additionally, methods that allow patients to prioritise items within the HOOS/KOOS should be explored, as such approaches may enhance their integration into shared decision-making. Furthermore, future studies should investigate the broader applicability of these findings across various healthcare settings. Lastly, these findings may not be confined to the HOOS and KOOS but likely apply to most condition-specific PROMs. This is supported by the limited use of other condition-specific PROMs for goal setting by physiotherapists in Sweden and their preference for the PCS [23], which aligns with our results. Therefore, future research should investigate physiotherapists’ experiences with other PROMs and their use in other conditions.

## 5. Conclusions

In conclusion, physiotherapists experience that the HOOS/KOOS-ADL and HOOS/KOOS-12 complement history taking and clinical examination, help to identify complications, and help to evaluate treatment progress and goals. However, the absence of normative reference scores in clinical practice guidelines and the lack of a feature to prioritise items for treatment goals appears to limit their use in value-based healthcare. To enhance their integration in daily clinical practice, there is a need for guidelines and policies that better align with the clinical context and electronic health record systems that support efficient PROM administration.

## Figures and Tables

**Table 1 jcm-14-00992-t001:** Participant demographics.

Participant Identifier	Participants’ Demographics	Employment Status	HOOS/KOOS Version	Yearly Treatment Volume	Interviewing Technique	Recruitment Strategy
P1	Female, age 25–29, with 4 years of experience, pursuing an MSc in geriatric physiotherapy	Employee	-Physical functio Short-form (5/7 items)	6–10 THA 6–10 TKA	Semi-structured interview	DAQP
P2	Male, age 45–49, 10 years of experience, BSc, Sports Physiotherapy	Employee	Original version (40/42 items)	6–10 THA 1–5 TKA	Semi-structured interview	DAQP
P3	Male, age 50–54, 25 years of experience, BSc, Manual Therapy	Practice owner	-Activities of the daily life subscale (17 items)	11–15 THA 11–15 TKA	Semi-structured interview	DAQP
P4	Male, age 30–34, with 8 years of experience, bachelor’s degree	Practice owner	-HOOS/KOOS-12 (12 items)	25+ THA 25+ TKA	Semi-structured interview	DAQP
P5	Female, age 35–39, 12 years of experience, MSc in Sport Physiotherapy	Employee, teacher	Original version (40/42 items)	1–5 THA 1–5 TKA	Semi-structured interview	Personal network
P6	Female, age 40–44, 17 years’ experience, BSc, Oedema therapist	Employee	-Activities of the daily life subscale (17 items)	1–5 THA 6–10 TKA	Semi-structured interview	Regional collaboration network
P7	Male, age 40–44, 11 years’ experience, BSc	Practice owner	-Physical function Short-form (5/7 items)	11–15 THA 16–20 TKA	Semi-structured interview	Regional collaboration network
P8	Male, age 25–29, 4 years’ experience, BSc	Employee	-Activities of the daily life subscale (17 items)	11–15 THA 16–20 TKA	Think-aloud	Regional collaboration network
P9	Female, age 25–29, 3 years’ experience, MSc Physiotherapy Sciences	Employee	-Physical function Short-form (5/7 items)	1–5 THA 11–15 TKA	Think-aloud	DAQP
P10	Female, age 25–29, 5 years’ experience, MSc Clinical Epidemiology	Employee	-Activities of the aily life subscale (17 items)	11–15 THA 16–20 TKA	Think-aloud	Regional collaboration network
P11	Male, age 30–34, 7 years, BSc	Employee	-Activities of the daily life subscale (17 items)	6–10 THA 6–10 TKA	Think-aloud	Regional collaboration network
P12	Female, age 25–29, 6 years’ experience, MSc Manual Therapy	Employee	-Activities of the daily life subscale (17 items)	16–20 THA 16–20 TKA	Think-aloud	Regional collaboration network
P13	Male, age 40–44, 21 years’ experience, MSc Sport Physiotherapy	Practice Owner	-Physical function Short-form (5/7 items)	1–5 THA 1–5 TKA	Think-aloud	DAQP

HOOS: Hip disability and Osteoarthritis Outcome Score, KOOS: Knee injury and Osteoarthritis Outcome Score, MSc: master of Science, BSc: bachelor of Science, THA: total hip arthroplasty, TKA: total knee arthroplasty, DAQP: Dutch Association for Quality in Physiotherapy.

**Table 2 jcm-14-00992-t002:** Overview of themes and subthemes regarding physiotherapist use of the HOOS and KOOS after THA and TKA.

Theme	Subthemes
1. Physiotherapists use the HOOS and KOOS for various purposes in daily clinical practice	*1.1* *HOOS and KOOS complement history taking and physical examination and identify complications* *1.2* *HOOS and KOOS help monitor progress and set treatment goals*
2. Perceptions of the relevance of the HOOS and KOOS for clinical practice vary per item, domain, and version	*2.1* *Items regarding ADL activities and psychosocial factors support clinical reasoning* *2.2* *Items of the pain, symptoms, and sports and recreation domains are unclear or irrelevant for most patients with THA or TKA* *2.3* The pain, *symptoms, and sport and recreation domains have little clinical value* *2.4* *Physiotherapists prefer the HOOS/KOOS-ADL and HOOS/KOOS-12 above the full version of the HOOS/KOOS and the HOOS/KOOS-PS*
3. Practical aspects of administration influence their use in daily clinical practice	*3.1* *Supportive electronic health record systems facilitate the use of HOOS and KOOS* *3.2* *Regulations regarding HOOS and KOOS administration imposed by insurance companies and professional associations have conflicting consequences* *3.3* *Shorter, more practical generic PROMs are preferred over the HOOS and KOOS*

HOOS: Hip disability and Osteoarthritis Outcome Score, KOOS: Knee injury and Osteoarthritis Outcome Score, -ADL: activities of daily living, -PS: physical functioning short-form, THA: total hip arthroplasty, TKA: total knee arthroplasty, PROMs: Patient Reported Outcome Measures.

## Data Availability

Codes supporting the conclusions of this article will be made available by the authors upon reasonable request.

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
