# Peer review of "Physiotherapists’ Experiences with the Hip Disability and Knee Injury Osteoarthritis Outcome Score Following Total Hip and Knee Arthroplasty: A Qualitative Interview Study"

_jcm, 2025, doi:10.3390/jcm14030992_

Round 1
Reviewer 1 Report
Comments and Suggestions for Authors
The authors did a good job defining and concluding the use of PROMs in practice, highlighting the problems of relying on a generic form for many medical issues.
Reviewer 2 Report
Comments and Suggestions for Authors
1. The Introduction mentions that physiotherapists’ experiences with HOOS/KOOS have not yet been explored, but it does not explain why this is important in detail. Additional explanations should be provided regarding the significance for clinical practice, patient outcome improvement, and the lack of qualitative research in this area.
2. It is recommended to begin the Discussion by restating the study's objectives.
3, Although the Discussion briefly addresses the need for shorter PROMs and EHR systems, it should emphasize actionable clinical implications. For example, revising guidelines to prioritize HOOS/KOOS-ADL and HOOS/KOOS-12 for THA/TKA patients, or recommending training programs for physiotherapists to enhance the effective use of PROMs in shared decision-making.
4. The limitations of the study should be described in more detail. For instance, the restricted geographic scope and its impact on generalizability, the influence of interviewers’ lack of experience on data collection and interpretation, and the potential effects of sampling on the representativeness of the results.
5. Future research directions should be described more concretely.
Reviewer 3 Report
Comments and Suggestions for Authors
In general, authors must review the scientific level of the paper. My comments are:
ABSTRACT:
· The abstract should be shorter, 200 words.
· Authors should explain in more detail what "experience" is.
· Authors should explain what the interview consisted of.
· The conclusion should be clearer and more concise. It should represent the main finding.
· Lines 41-43 must be deleted
INTRODUCTION
· There is an error in the references on line 50, 61, 64,… Checking all text.
· The objective at the end of the introduction should be clear. Usually, it is the last paragraph. Authors should further describe the objective of the study.
· MATERIAL AND METHODS
· Who interprets the interview responses? One person or were there two who contracted the ideas?
· The final sample was 13 interviews? Is this a sufficient sample size? The authors calculated the sample size.
· What is the difference between a semi-structured interview and Think-Aloud? How was the selection made? Random?
· RESULTS
· The results is too extensive. Authors should make the effort to detail the important aspects of the results.
· The figures are not sharp. They need to be improved.
· DISCUSSION
Authors must check the verb tense of the entire text. There are sentences that are written in the present and are in the past.
REFERENCES
· The style of the references is not correct
Round 2
Reviewer 2 Report
Comments and Suggestions for Authors
The authors all modified it appropriately.
Reviewer 3 Report
Comments and Suggestions for Authors
The authors have responded to my comments appropriately.